# Integrated Metabolites and Transcriptomics at Different Growth Stages Reveal Polysaccharide and Flavonoid Biosynthesis in *Cynomorium songaricum*

**DOI:** 10.3390/ijms231810675

**Published:** 2022-09-14

**Authors:** Jie Wang, Hongyan Su, Zhibo Wu, Wenshu Wang, Yubi Zhou, Mengfei Li

**Affiliations:** 1Qinghai Key Laboratory of Qinghai-Tibet Plateau Biological Resource, Northwest Institute of Plateau Biology, Chinese Academy of Sciences, Xining 810008, China; 2State Key Laboratory of Aridland Crop Science, College of Life Science and Technology, Gansu Agricultural University, Lanzhou 730070, China; 3Station of Alxa League Aviation Forest Guard, Alxa 750306, China; 4Alxa Forestry and Grassland Research Institute, Alxa 750306, China

**Keywords:** *Cynomorium songaricum*, polysaccharide biosynthesis, flavonoid biosynthesis, different growth stages, transcriptomics, gene expression

## Abstract

*Cynomorium songaricum* is a perennial parasitic herb, and its stem is widely used as a traditional Chinese medicine, which largely relies on bioactive compounds (e.g., polysaccharides, flavonoids, and triterpenes). To date, although the optimum harvest time of stems has been demonstrated at the unearthed stage (namely the early flowering stage, EFS), the accumulation mechanism of polysaccharides and flavonoids during growth stages is still limited. In this study, the physiological characteristics (stem fresh weight, contents of soluble sugar and flavonoids, and antioxidant capacity) at four different growth stages (germination stage (GS), vegetative growth stage (VGS), EFS, and flowering stage (FS)) were determined, transcriptomics were analyzed by illumina sequencing, and expression levels of key genes were validated by qRT-PCR at the GS, VGS, and EFS. The results show that the stem biomass, soluble sugar and total flavonoids contents, and antioxidant capacity peaked at EFS compared with GS, VGS, and FS. A total of 6098 and 13,023 differentially expressed genes (DEGs) were observed at VGS and EFS vs. GS, respectively, with 367 genes co-expressed. Based on their biological functions, 109 genes were directly involved in polysaccharide and flavonoid biosynthesis as well as growth and development. The expression levels of key genes involved in polysaccharides (e.g., *GLCs*, *XTHs* and *PMEs*), flavonoids (e.g., *4CLLs*, *CYPs* and *UGTs*), growth and development (e.g., *AC58*, *TCPs* and *AP1*), hormones biosynthesis and signaling (e.g., *YUC8*, *AIPT* and *ACO1*), and transcription factors (e.g., *MYBs*, *bHLHs* and *WRKYs*) were in accordance with changes of physiological characteristics. The combinational analysis of metabolites with transcriptomics provides insight into the mechanism of polysaccharide and flavonoid biosynthesis in *C. songaricum* during growth stages.

## 1. Introduction

*Cynomorium songaricum* Rupr. is a perennial parasitic herb that is primarily distributed in the desert and saline areas of northwest China, including: Qinghai, Xinjiang, and Inner Mongolia [1]. The stem of *C. songaricum* is a traditional Chinese medicine and widely used to tonify kidneys, replenish essence, supplement the blood, and relax the bowels [2]. In recent years, the stem has also been applied in anti-oxidation, anti-viral and anti-obesity diseases, which largely rely on bioactive compounds such as polysaccharides, flavonoids, and triterpenes [3,4,5,6]. 

In order to improve the yield and quality of *C. songaricum*, several studies have been conducted on different host plants and harvest stages. Previous studies have found that *C. songaricum* is mainly a host in four species, including: *Nitraia*
*roborowskii* Kom., *N.*
*sibirica* Pall., *N.*
*tangutorum* Bobr., and *N.*
*sphaerocarpa* Maxim [7,8,9]; additionally, there were greater stem biomass and polysaccharides content in the *C. songaricum* host in *N. roborowskii* than *N. sibirica* [10]. The types and contents of liposoluble components (e.g., β-sitosterol, hexadecanoic acid, and (Z)-9-octadecenoic acid) in *C. songaricum* were significantly different among different hosts [11]. For other parasitic plants, there were significant differences in contents of flavonoids (e.g., rutin, isorhamnetin, and kaempferol) in *Cuscuta chinensis* from different hosts [12]. Generally, the optimum harvest time of the *C. songaricum* stem is between the unearthed prophase (namely vegetative growth stage, VGS) and unearthed stage (namely early flowering stage, EFS), with greater contents of polysaccharides and protocatechuic acid compared with the flowering stage (FS) and fruiting stage [13,14]. Extensive studies have demonstrated that reproductive growth significantly decreased the accumulation of bioactive compounds. For example, the contents of ferulic acid and total flavonoids in the root of *Angelica sinensis* decreased with prolonged bolting and flowering [15]; the contents of hypericin and total flavonoids and polyphenols in aerial parts of *Hypericum perforatum* on a per plant basis maximized at the EFS, while significantly decreased at the FS and fruiting stage [16]; the nutrients (water-soluble carbohydrate and crude protein) and the silage quality of *Lolium multiflorum* were higher and better at the earring stage than the booting and flowering stages [17]. At the molecular level, to reveal the genes regulating bioactive compounds, two genes, leucoanthocyanidin reductase (*LAR*) and dihydroflavonol 4-reductase (*DFR*), have been identified to be involved in catechin biosynthesis based on transcriptome analysis of *C. Songaricum* [18]. 

As known, polysaccharides are critical quality markers for many medicinal plants and exhibit immunomodulatory, anti-diabetic, anti-aging, anti-cancer, and anti-oxidant properties [19]. Plant polysaccharides are composed of a variety of monosaccharides with complex structure [20]. Their biosynthetic pathway mainly includes three parts: (1) sucrose is converted to Glc-1P and Fru-6P; (2) uridine diphosphate glucose (UDP-Glc) is derived from Glc-1P immediately, and Fru-6P is converted to GDP-Man indirectly, meanwhile, other nucleotide-diphospho-sugar (NDP) sugars are further converted via the NDP-sugar interconversion enzymes (NSEs); and (3) various NDP-sugars form growing polysaccharide chains via the glycosyltransferases (GTs) [21,22,23]. During these processes, many key enzymes are involved such as sucrose synthase (SUS), sucrose phosphate synthase (SPS), invertase (INV), hexokinase (HK), fructokinase (FRK), UDP-glucose hydrogenase (UGDH), UDP-glucose pyrophosphorylase (UGPase), UDP-glucose rhamnose synthase (RHM), phosphomannose isomerase (PMM), GDP-mannose pyrophosphorylase (GMPP), and GTs [19,20,21]. In addition, polysaccharide biosynthesis is found to be regulated by the MYB transcription factors (TF) [24].

Flavonoids are generally classified into seven subclasses including: flavonols, flavones, isoflavones, anthocyanidins, flavanones, flavanols, and chalcones, which exhibit anti-inflammatory, anti-cancer, and anti-oxidant properties and reduce the risk of cardiovascular disease [25]. Flavonoids are derived from *p*-coumaroyl CoA (via the shikimate pathway) and malonyl CoA (via the acetate pathway) [26]. *p*-coumaroyl CoA is synthesized from phenylalanine by the catalyzation of phenylalanine ammonialyase (PAL), cinnamic acid hydroxylase (C4H), and coumarin CoA ligase (4CL) [27,28]. Naringenin chalcone is produced from *p*-coumaroyl CoA and malonyl CoA by the condensation and isomerization of chalcone synthase (CHS) [29]. Then, flavanones and naringenin are produced by the catalyzation of chalcone isomerase (CHI) [26]. Naringenin is the common precursor of most intermediate metabolites and end products in the synthesis of flavonoids. Flavones can be produced from naringenin via the flavone synthase I (FNS I) or flavone synthase II (FNS II), isoflavones can be produced via isoflavone synthase (IFS), and dihydrokaempferol can be produced via the flavanone-3-hydroxylase (F3H). Meanwhile, dihydroquercetin and dihydromyricetin can be produced via flavonol 3′-hydroxylase (F3′H) and flavonol 3′5′-hydroxylase (F3′5′H), respectively. Flavonols (e.g., kaempferol, quercetin and myricetin) and leucoanthocyanidins can be synthesized from dihydroflavonol under the action of flavonol synthase (FLS) and DFR, respectively. Subsequently, anthocyanidins can be synthesized by leucoanthocyanidin dioxygenase (LDOX), and then anthocyanins (e.g., pelargonin, cyandin, and delphinidin) can be produced by anthocyanidin synthase (ANS) and uridine diphosphate (UDP)-glucose flavonoid-3-*O*-glycosyltransferase (UFGT); furthermore, leucoanthocyanidins can generate flavanols via LAR, producing proanthocyanins [30,31]. Generally, flavonoid biosynthesis is regulated by the TFs such as MYB, bZIP, bHLH or their complex [32,33]; in addition, cytochrome P450s (CYPs) are also observed to be involved in flavonoid biosynthesis [34]. Finally, these relatively stable flavonoids that are modified through acylation (e.g., acetyltransferase, ATs), methylation (e.g., methyltransferases, MTs), and glycosylation (e.g., UDP-glycosyltransferases, UGTs) can accumulate in plants [35,36].

To date, although the host plants and optimum harvest time of stem at the EFS, based on the stem biomass and metabolites content, have been demonstrated in previous studies [7,8,9,10,13,14], the accumulation mechanism of stem biomass, polysaccharides, and total flavonoids of *C. songaricum* during growth stages has not been revealed. In this study, the physiological characteristics (e.g., fresh weight, contents of soluble sugar and flavonoids, and antioxidant capacity) of *C. songaricum* at four different growth stages (germination stage (GS), vegetative growth stage (VGS), early flowering stage (EFS), and flowering stage (FS)) were determined, the transcriptomics were analyzed, and expression levels of key genes were validated at the GS, VGS, and EFS. We found that there were significant differences in stem biomass, contents of soluble sugar and flavonoids, antioxidant capacity, as well as expression levels of genes involved in plant growth and polysaccharide and flavonoid biosynthesis in *C. songaricum* during the growth stages.

## 2. Results

### 2.1. Changes of Fresh Weight at Different Growth Stages

As shown in Figure 1, there is a significant difference in the stem fresh weight (FW) between the four different growth stages, with a 3.85- and 1.32-fold increase at the EFS compared with the GS and VGS, respectively, while there is a 1.22-fold decrease at the FS compared with the EFS.

### 2.2. Changes of Soluble Sugar and Total Flavonoids Contents at Different Growth Stages

Significant differences in soluble sugar and total flavonoids contents in stems were observed at the four different growth stages (Figure 2). The soluble sugar and total flavonoids contents showed a similar change trend, with a decrease at the VGS, an increase at the EFS, and then a decrease at the FS. 

### 2.3. Changes of Antioxidant Capacity at Different Growth Stages

Significant differences in antioxidant capacities of extracts from stems were observed at the four different growth stages (Figure 3). The DPPH scavenging activity and FRAP value showed a similar change trend, with a decrease at the VGS, an increase at the EFS, and then a decrease at the FS. 

### 2.4. Transcriptomics Analysis at Different Growth Stages

#### 2.4.1. Global Gene Analysis

To reveal the accumulation mechanism of stem biomass, soluble sugar, and total flavonoids during the growth stages of *C. songaricum*, a comparison of the transcripts was performed between the GS, VGS, and EFS. After data filtering, 50.22, 51.24, and 52.69 million high-quality reads were collected, and 41.17, 42.46. and 43.59 million unique reads with 1.62, 1.61. and 1.62 million multiple reads were mapped at the GS, VGS, and EFS, respectively (Table 1; Appendix A).

A total of 95,126 unigenes were annotated on KEGG (10,274), KOG (17,550), Nr (40,427), and Swissprot (16,181) databases (Appendix A). Using the KEGG database, 6098 DEGs at VGS vs. GS were enriched for 103 metabolism pathways such as global and overview maps, energy metabolism, and carbohydrate metabolism; 13,023 DEGs at EFS vs. GS were enriched for 123 metabolism pathways such as global and overview maps, carbohydrate metabolism, and energy metabolism (Appendix A). Using the KOG database, 18.45% of unigenes encoded the identified proteins that could be classified into 25 functional categories (Appendix A). Using the NR database, the top 10 species include: *Cajanus cajan*, *Vitis vinifera*, *Cephalotus follicularis*, *Theobroma cacao*, *Nicotiana attenuata*, *Juglans regia*, *Corchorus capsularis*, *Brassica napus*, *B. rapa*, and *Medicago truncatula* (Appendix A). Using the SwissProt database, 17.01% of unigenes were annotated. Using the GO database, the DEGs were classified into three ontologies, including biological process (BP), cellular component (CC), and molecular function (MF) (Appendix A).

#### 2.4.2. Identification of Differentially Expressed Genes (DEGs)

A total of 6098 and 13,023 DEGs were observed from 95,126 unigenes, with 3398 up-regulated (UR) and 2700 down-regulated (DR) at VGS vs. GS, and 4516 UR and 8507 DR at EFS vs. GS (Figure 4), based on the principal component analysis (Appendix A) and Pearson correlation analysis (Appendix A). 

#### 2.4.3. Distribution and Classification of DEGs 

Among the 6098 and 13,023 DEGs at VGS vs. GS and EFS vs. GS, 950 and 2847 genes were identified from the KEGG, KOG, Swiss-Prot or GO databases, respectively (Figure 5A,B). Of the identified DEGs, 367 genes co-expressed at the GS, VGS, and EFS (Figure 5C). Based on their biological functions, the 367 genes were classified into 10 categories, including primary metabolism (79), secondary metabolism (16), cell morphogenesis (28), bio-signaling (41), transcription factor (38), transport (39), photosynthesis and energy (43), polynucleotide metabolism (29), protein metabolism (23), and stress response (31) (Figure 5D).

### 2.5. Functional Classification of DEGs

#### 2.5.1. DEGs Involved in Polysaccharide Biosynthesis

Among the 79 DEGs associated with primary metabolism, 24 genes were identified to be involved in polysaccharide biosynthesis, including: glucose (*BGLU23*, *BGLU44*, *At5g56590*, *GAPA*, and *MTH_209*), sucrose (*INVA* and *BFRUCT3*), fructose (*FBA2*, *PFP-BETA*, and *At3g55800*), xylan (*BXL5*, *GT17*, *TBL19*, *TBL31*, *ESK1*, and *XTH9*), trehalose (*TPPF*), and pectin (*PAE8*, *GAUT12*, *At5g63180*, *PME7*, *PMEI10*, *PME15,* and *PME40*) (Appendix A). The other 55 genes were involved in lipid, fatty acid, and amino acid metabolism (Appendix A). The expression levels of 12 select genes involved in polysaccharide biosynthesis were validated by qRT-PCR, with a 1.67- (*INVA*) to 21.08-fold (*BGLU2**3*) UR of the 12 genes, while there was a 0.41-fold (*PMEI10*) DR at VGS and EFS vs. GS (Figure 6). Meanwhile, the relative expression levels (RELs) were consistent with their Reads Per kb per Million (RPKM) values (Appendix A).

#### 2.5.2. DEGs Involved in Flavonoid Biosynthesis

Among the 16 DEGs associated with secondary metabolism, 11 genes were identified to be involved in flavonoid biosynthesis including: *4CLL1*, *4CLL6*, *HST*, *CHI3*, *CAD9*, *CYP714C2*, *CYP93B1*, *F6′H1*, *UGT84A13*, *UGT87A1*, and *UGT94E5* (Appendix A). The other 5 genes were involved in terpene biosynthesis (Appendix A). The expression levels of the 11 genes involved in flavonoid biosynthesis were validated, with a 1.96-(*UGT94E5*) to 8.81-fold (*4CLL6*) UR of 10 genes, while there was a 0.57-fold (*UGT87A1*) DR at VGS vs. GS; and a 1.84- (*CHI3*) to 13.35-fold (*4CLL6*) UR of 7 genes, while there was a 0.52-(*UGT87A1*) to 0.99-fold (*F6′H1*) DR of 4 genes at EFS vs. GS (Figure 7). Meanwhile, the RELs were consistent with their RPKM values (Appendix A).

#### 2.5.3. DEGs Involved in Cell Growth and Flower Development 

Among the 28 DEGs associated with cell morphogenesis, 20 genes were identified to be involved in cell growth (9 genes; *AC58*, *ADF*, *ATJ11*, *CYCP3-1*, *SDS*, *LRX6*, *MIZ1*, *PATROL1*, and *TBB7*) and flower development (11 genes; *CSLD4*, *EXLB1*, *AP1*, *ASOL*, *AMP1*, *TKPR2*, *TCTP1*, *CYP704B1*, *HAT*, *Os05g0239150*, and *Os05g0583200*) (Appendix A). The other eight genes were involved in other cell morphogenesis, such as seed development and programmed cell death (Appendix A). The expression levels of eight select genes involved in cell growth and flower development were validated, with five genes showing a 2.85-(*CYCP3-1*) to 17.98-fold (*AC58*) UR, while three genes showed a 0.43-(*HAT*) to 0.91-fold (*AMP1*) DR at VGS and EFS vs. GS (Figure 8). Meanwhile, the RELs were consistent with their RPKM values (Appendix A).

#### 2.5.4. DEGs Involved in Hormone Biosynthesis and Signaling

Among the 41 DEGs associated with bio-signaling, 22 genes were identified to be involved in hormone biosynthesis, including: auxin (*YUC8*), cytokinin (*AIPT* and *LOG5*), gibberellin (*LE*), ethylene (*ACO1* and *ACS1*), and abscisic acid (*CYP707A6*), and hormone signaling, including: auxin (*AUX22D*, *ARP12.5*, *SAUR71*, *GH3.6*, and *PILS2*), cytokinin (*AHK4*), jasmonic acid (*JOX2*), and ethylene (*AIL1*, *AIL6*, *ETR2*, *SHN3*, *REF6*, *ERF010*, *ERF034*, and *ERF114*) (Appendix A). The other 19 genes were involved in other bio-signaling, such as protein kinase and calcium and receptor serine/threonine kinase (Appendix A). The expression levels of 15 select genes were validated, with genes involved in auxin (e.g., *YUC8*, *AUX22D*, and *GH3.6*) and cytokinin (e.g., *AIPT*, *LOG5*, and *AHK4*) showing UR, while in abscisic acid (e.g., *CYP707A6*) and ethylene (e.g., *ACO1*, *AIL1* and *ERF010*) showing DR at VGS and EFS vs. GS (Figure 9). Meanwhile, the RELs were almost consistent with their RPKM values (Appendix A).

#### 2.5.5. TFs Involved in Flavonoid Biosynthesis as Well as Growth and Development

Among the 38 DEGs associated with TFs, 28 TFs were identified to be involved in flavonoid biosynthesis, including: MYB (6 genes; *LIMYB*, *MYB2*, *MYB14*, *MYB52*, *MYB83*, and *MYB306*), bHLH (2 genes; *BHLH52* and *BHLH94*), and WRKY (4 genes; *WRKY6*, *WRKY53*, *WRKY70* and *WRKY72*), as well as growth and development, including: cell growth (8 genes; *DREB2D*, *DREB3*, *At2g01810*, *MMD1*, *PRE5*, *TCP9*, *TCP18*, and *UPB1*) and flower development (8 genes; *AHL17*, *AHL20*, *AHL22*, *AHL23*, *MIP1B*, *HEC2*, *SCRM*, and *PAN*) (Appendix A). The other 10 genes were involved in other TFs such as Transcription repressor OFPs, B3 domain-containing proteins, and Zinc finger CCCH domain-containing proteins (Appendix A). The expression levels of 12 select genes involved in flavonoid biosynthesis as well as growth and development were validated, with most TFs (e.g., *MYB83*, *BHLH52*, *WRKY53*, *TCP9*, and *UPB1*) showing UR, while some TFs (e.g., *MYB2*, *WRKY72*, *At2g01810*, and *AHL20*) showed DR at VGS and EFS vs. GS (Figure 10). Meanwhile, the RELs were almost consistent with their RPKM values (Appendix A).

#### 2.5.6. DEGs Involved in Polysaccharide Transport

Among the 39 DEGs associated with transport, 6 genes were identified to be involved in polysaccharide transport, including: sugar transporter (*SWEET5*, *slc37a2* and *At5g55950*) and ABC transporter family (*ABCB2*, *ABCG1*, and *ABCG22*) (Appendix A). The other 33 genes were involved in other transport, such as protein, lipid, and amino acid (Appendix A). The expression levels of the six genes involved in polysaccharide transport were validated, with four genes (*SWEET5*, *slc37a2*, *At5g55950*, and *ABCB2*) showing UR, while two genes (*ABCG1* and *ABCG22*) showed DR at VGS and EFS vs. GS (Figure 11). Meanwhile, the RELs were almost consistent with their RPKM values (Appendix A).

#### 2.5.7. DEGs Involved in Other Biological Functions

In this study, there are 126 DEGs involved in other biological functions, including: photosynthesis and energy (43 genes; Appendix A), polynucleotide metabolism (29 genes; Appendix A), protein metabolism (23 genes; Appendix A), and stress response (31 genes; Appendix A). These genes may also participate in the biosynthesis of polysaccharides and flavonoids as well as the growth and development of *C*. *songaricum*.

## 3. Discussion

Although the optimum harvest time of the *C*. *songaricum* stem at EFS has been demonstrated in previous studies [12,13], the mechanism of polysaccharide and flavonoid biosynthesis during growth stages is still limited. In this study, the maximum values of stem fresh weight, soluble sugar and total flavonoids contents, and antioxidant capacity in *C*. *songaricum* were observed at EFS compared with GS, VGS, and FS; a total of 367 DEGs co-expressed at the GS, VGS, and EFS with 109 genes directly involved in polysaccharide and flavonoid biosynthesis as well as growth and development.

Previous studies have found that there was a significant decrease in bioactive compounds (polysaccharides and protocatechuic acid) in *C. songaricum* during reproductive growth [12,13]. Here, we also found that there was a significant increase in stem biomass and content of bioactive compounds (polysaccharides and flavonoids) during vegetative growth (from the GS to EFS), while significant decrease during reproductive growth (from the EFS to FS). Extensive studies have demonstrated that there is a significant positive relationship between antioxidant capacity and bioactive compounds (e.g., polysaccharides, flavonoids, and phenols) in plants [37,38,39]. Here, the antioxidant capacities (DPPH scavenging activity and FRAP value) also showed a similar change trend with the contents of soluble sugar and total flavonoids, which indicates that the constituents of polysaccharides and flavonoids play critical roles in the pharmacological properties of *C. songaricum*.

Polysaccharide biosynthesis is a complex process due to its various structures and involvement of many enzymes (e.g., SUS, SPS, INV, UGDH, UGPase, and GTs) in the polysaccharide metabolic pathways [20,21,22,23]. Here, 24 genes involved in polysaccharide biosynthesis participate in the metabolism of glucose, fructose, xylan, trehalose, and pectin. For example, BGLU23 and GAPA can hydrolyze the 1,3-beta-D-glucosidic linkages in 1,3-beta-D-glucans to produce beta-D-glucose [40]; INVA is involved in sucrose metabolism by the hydrolysis of the terminal beta-D-fructofuranoside residues in beta-D-fructofuranosides [41]; FBA2 plays a key role in glycolysis that is part of carbohydrate degradation [42], while At3g55800 is involved in the pathway calvin cycle that is part of carbohydrate biosynthesis [43]; BXL5 is involved in the xylan catabolic process, while TBL31 in the xylan biosynthetic process [44]; XTH9 participates in cleaving xyloglucan polymers [45]; and PAE8 and PME7 are involved in pectin degradation, while the PMEI10 inhibits the pectin degradation [46,47,48]. These molecular functions of select genes once again prove that the polysaccharide metabolic process is complex and regulated by multi-genes and multi-pathways. In addition, three genes may be involved in polysaccharide transport, with SWEET5 participating in sugar transmembrane transporter activity [49], slc37a2 in transporting cytoplasmic glucose-6-phosphate into the lumen of the endoplasmic reticulum [50]; and At5g55950 in nucleotide-sugar transmembrane transport [51].

The flavonoid synthesis is also complex in plants, and their metabolic pathways are generally regulated by a series of key enzymes, including: PAL, C4H, 4CL, CHS, CHI, F3′H, and UGTs [26,27,28,29,30,31]. In this study, 11 genes were involved in flavonoid biosynthesis, including: *4CLL1*, *4CLL6*, *HST*, *CHI3*, *CAD9*, *CYP714C2*, *CYP93B1*, *F6′H1*, *UGT84A13*, *UGT87A1*, and *UGT94E5*. Clearly, 4CLL1, 4CLL6, CHI3, F6′H1, UGT84A13, UGT87A1, and UGT94E5 directly participate in flavonoid biosynthesis. CYP714C2 and CYP93B1 may also participate in flavonoid biosynthesis, with the overexpression of the two *CYP714C2* and *CYP93B1* genes enhancing flavonoid accumulation [52]. In addition, HST and CAD9 are the key branch enzymes in the phenylpropanoid metabolic pathway, leading to lignin/lignan biosynthesis [53,54]. Generally, flavonoids can be transported through proton antiport and ABC-type transporter in plants [55]. In this study, three ABC transporters (*ABCB2*, *ABCG1*, and *ABCG22*) were observed to be involved in flavonoid transport [56].

Although *C*. *songaricum* is a parasitic plant, growth and development are essential for plant morphogenesis. As described previously, stem quality (i.e., biomass and bioactive compound contents) was significantly affected by the growth stages. In this study, 20 genes were involved in cell growth and flower development. Examples for the cell growth: AC58 plays an important role in cell shape determination and cell division [57]; CYCP3-1 can regulate meristem cell division and lateral root development [58]; and MIZ1 participates in lateral root development [59]. For flower development, AP1 and AMP1 are involved in flower development [60,61]; TKPR2 is involved in pollen exine formation [62]; and HAT is essential for plant growth and development, especially in post-embryonic development [63].

It is noteworthy that endogenous hormones play critical roles in cell growth and flower development. In this study, 22 genes were involved in hormone biosynthesis and signaling. Specifically, in the hormone biosynthesis, YUC8 is involved in auxin biosynthesis [64]; AIPT and LOG5 are involved in cytokinin biosynthesis [65,66]; LE is involved in gibberellin biosynthesis by converting the inactive GA_9_ and GA_20_ in the bioactives GA_4_ and GA_1_ [67]; ACO1 and ACS1 are involved in ethylene biosynthesis [68,69]; and CYP707A6 is involved in the oxidative degradation of abscisic acid [70]. For bio-signaling, AUX22D, SAUR71, and GH3.6 regulate cell expansion, root cell differentiation, and shoot cell elongation by mediating the auxin-activated signaling pathway [71,72,73]; AHK4 regulates many developmental processes such as cell division, root repression, and shoot promotion by acting as a positive regulator of cytokinin signaling [74]; and AIL1, ERF010, and ERF114 are involved in cell proliferation and axillary bud outgrowth by acting as a transcriptional regulator or integrator of ethylene signaling [75,76].

Moreover, TFs play vital roles in the biosynthesis of polysaccharides and flavonoids as well as plant growth and development [77]. Previous studies have demonstrated that MYBs, BHLHs, and WRKYs or their complex (e.g., MYB-bHLH) play regulatory roles in flavonoid biosynthesis [78,79,80]. In this study, MYBs (e.g., MYB2, MYB14, and MYB83), BHLHs (e.g., BHLH52 and BHLH94), and WRKYs (e.g., WRKY6, WRKY53, and WRKY72) may play an important role in regulating flavonoid biosynthesis in *C*. *songaricum*. Meanwhile, MYBs may also participate in polysaccharide biosynthesis [24]. In addition, 16 TFs were involved in plant growth and development. For example, TCP9 and TCP18 may participate in axillary bud and root development [81,82]; UPB1 can modulate the balance between cellular proliferation and differentiation in root growth [83]; AHL20 acts as a negative regulator of FLOWERING LOCUS T (FT) that is a downstream floral integrator [84]; and BHLH52 may be related to floral organ development [85].

## 4. Materials and Methods

### 4.1. Plant Materials

Stems of *C. songaricum* host in *N. roborowskii* were collected at four different growth stages: GS, VGS, EFS, and FS on 15 March, 15 April, 15 May, and 15 June 2019, respectively, from Dulan county (2800 m; 36°2′25″ N, 97°40′26″ E) of Qinghai, China (Figure 12). The stems were cleaned and rapidly frozen in liquid nitrogen, the middle part of the stem was used for the determination of soluble sugar and total flavonoids contents as well as antioxidant capacity, and the shoot apical meristems (SAM) were used for transcriptomic analysis.

### 4.2. Measurement of Stem Biomass

Stem fresh weight (FW) was immediately measured after *C. songaricum* was dug out from the soil. Specifically, 10 sites of *C**. songaricum* were randomly selected from 10 host plants of *N. roborowskii*, onevmedium-length stem was chosen from the independent site of *C**. songaricum* and measured using an electronic balance, and then the average FW of the 10 stems was calculated. Generally, the *C**. songaricum* plants grow in clusters, and the sizes of stems are basically the same (Appendix A).

### 4.3. Determination of Soluble Sugar and Total Flavonoids Contents as Well as Antioxidant Capacity

#### 4.3.1. Extracts Preparation

Extracts were prepared according to previous protocols [10]. Briefly, fresh stems (1.0 g) were ground into homogenate by adding ethanol (95% *v*/*v*, 20 mL), agitated at 120 r/min and 22 °C for 72 h, then centrifuged (TGL20M, Changsha, China) at 5000 r/min and 4 °C for 10 min. The supernatant was increased by 20 mL with ethanol (95% *v*/*v*), and then kept at 4 °C for the determination of soluble sugar, flavonoids, and antioxidant capacity.

#### 4.3.2. Determination of Soluble Sugar Content

Soluble sugar content was determined using the phenol-sulfuric acid method [10,86]. Briefly, extracts (15 μL) were added in the reaction. An absorbance reader was taken at 485 nm using a spectrometer (V1800, Shanghai, China). Soluble sugar content was calculated based on mg of sucrose.

#### 4.3.3. Determination of Flavonoids Content

Flavonoids content was determined using the NaNO_2_-AlCl_3_-NaOH method [87,88]. Briefly, the extracts (80 μL) were added into ddH_2_O (2 mL) and NaNO_2_ (5% *w*/*v*, 0.3 mL); after oscillation, AlCl_3_ (10% *w*/*v*, 0.3 mL) was added and reacted at 22 °C for 1 min; then, NaOH (1.0 mol/L, 2 mL) was added to stop the reaction. An absorbance reader was taken at 510 nm using a spectrometer (V1800, Shanghai, China). Flavonoid content was calculated based on milligram of catechin.

#### 4.3.4. Determination of Antioxidant Capacity

Antioxidant capacity was determined using two different methods: 1,1-diphenyl-1-picrylhydrazyl (DPPH) and ferric reducing antioxidant power (FRAP) [37,89].

DPPH radical scavenging assay was determined according to a previous protocol [90,91]. Briefly, extracts (5 μL) were added in the reaction. An absorbance reader was taken at 515 nm using a spectrometer (V1800, Shanghai, China). The capacity to scavenge DPPH radicals was calculated as follows:DPPH scavenging activity (%) = [(*A*_0_ − *A*)/*A*_0_] × 100
where “*A*_0_” and “*A*” were the absorbance of DPPH without and with sample, respectively.

FRAP assay was determined according to a previous protocol [91,92]. Briefly, extracts (10 μL) were added in the reaction. An absorbance reader was taken at 593 nm using a spectrometer (V1800, Shanghai, China). The FRAP value was calculated on the basis of (FeSO_4_·7H_2_O, 500 μmol Fe (II)/g), as follows:FRAP value (µmol Fe(II)/g) = [(*A* − *A*_0_)/(*A_FeSO_*_4*·*7*H*2*O*_ − *A*_0_)] × 500 (µmol Fe(II)/g)
where “*A*_0_” and “*A*” are the absorbance of FRAP without and with sample, respectively; *A_FeSO_*_4*·*7*H*2*O*_ is the absorbance of FeSO_4_·7H_2_O.

### 4.4. Transcriptomic Analysis

#### 4.4.1. RNA Extraction and Illumina Sequencing

Total RNA samples at GS, VGS, and EFS with three biological replicates were extracted using an RNA kit (R6827, Omega Bio-Tek, Inc., Norcross, GA, USA) according to the manufacturer’s protocols. The quality of the total RNA was determined using an Agilent 2100 Bioanalyzer (Agilent Technologies, Palo Alto, CA, USA). The processes of enrichment, fragmentation, reverse transcription, synthesis of the second-strand cDNA, and purification of cDNA fragments were applied according to previous protocols [93]. Reads were generated by using an Illumina HiSeqTM 4000 platform (Gene Denovo Biotechnology Co., Ltd., Guangzhou, China).

#### 4.4.2. Reads Filtration, Assembly, Unigene Expression Analysis, and Basic Annotation

Raw reads were filtered using a FASTQ system to obtain high-quality clean reads by removing reads containing adapters, removing reads containing more than 10% of unknown nucleotides (N), and removing low-quality reads containing more than 50% low-quality (Q-value ≤ 20) bases [94]. Clean reads were assembled using Trinity [95]. The expression level of each transcript was normalized to the values of the Reads Per kb per Million (RPKM). Differential expression analysis of transcripts was performed using DESeq2 software between different groups [96]. The differential expression levels between VGS vs. GS and EFS vs. GS were determined with the criteria of the false discovery rate (FDR) < 0.05 and |log_2_(fold-change)| > 1. The function of DEGs was annotated using BLAST against the databases, including Nr, KEGG, KOG, Swiss-Prot, and GO with e-value ≤ 10^−5^ as a threshold [97].

### 4.5. qRT-PCR Validation

The primer sequence (Appendix A) was designed via a primer-blast in NCBI and synthesized by Sangon Biotech Co., Ltd. (Shanghai, China). First, cDNA was synthesized using a RT Kit (KR116, Tiangen, China). PCR amplification was performed using a SuperReal PreMix (FP205, Tiangen, China). Melting curve was analyzed at 72 °C for 34 s. The *Actin* gene was used as a reference control [10]. The RELs of genes were calculated using a 2^−^^∆∆Ct^ method [98].

### 4.6. Statistical Analysis

All the measurements were performed using three biological replicates. Duncan tests of SPSS 22.0 software was used for statistical comparisons, with *p* < 0.05 considered significant.

## 5. Conclusions

From the above observations, the accumulation of polysaccharides and flavonoids reached the highest levels at the EFS during growth stages. A total of 6098 and 13,023 DEGs were observed at the VGS and EFS vs. GS, respectively, with 109 genes directly involved in polysaccharide and flavonoid biosynthesis as well as growth and development. The specific roles of key genes in the regulation of polysaccharide (e.g., *GLCs, XTHs*, and *PMEs*) and flavonoid (e.g., *4CLLs*, *CYPs* and *UGTs*) biosynthesis will require additional studies. These findings will provide theoretical and useful information for the large-scale cultivation and collection of *C. songaricum*, as well as improve the yield and quality of *C. songaricum* by regulating the nutrient transport of the host *Nitraia* species.

## Figures and Tables

**Figure 1 ijms-23-10675-f001:**
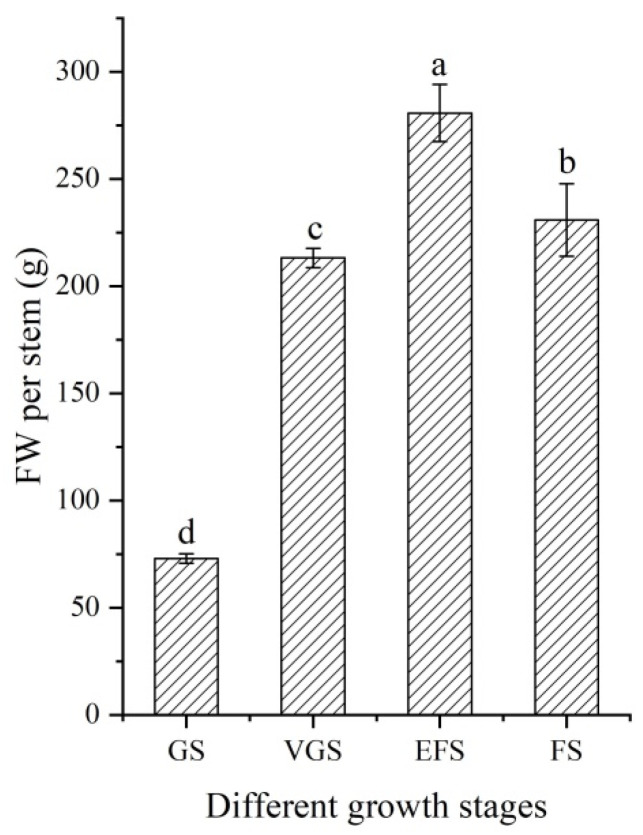
Stem fresh weight of *C. songaricum* at four different growth stages (mean ± SD, n = 10). Abbreviations: GS, germination stage; VGS, vegetative growth stage; EFS, early flowering stage; FS, flowering stage. Different letters represent a significant difference (*p* < 0.05) at different growth stages.

**Figure 2 ijms-23-10675-f002:**
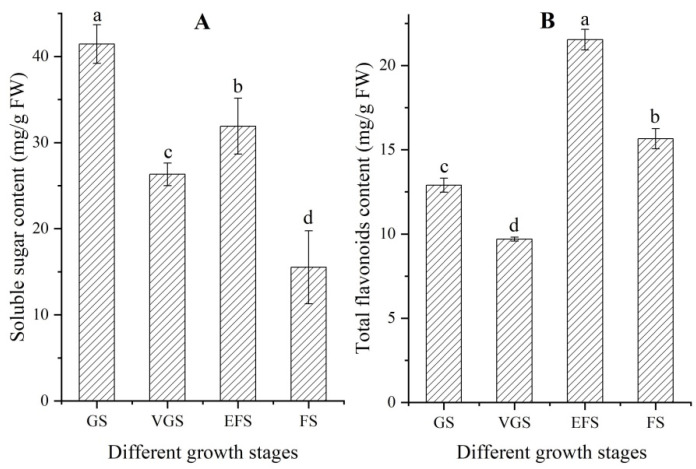
Soluble sugar and total flavonoids contents in stems of *C. songaricum* at four different growth stages (mean ± SD, n = 3). Images (**A**,**B**) represent soluble sugar and total flavonoids contents, respectively. Different letters represent a significant difference (*p* < 0.05) at different growth stages.

**Figure 3 ijms-23-10675-f003:**
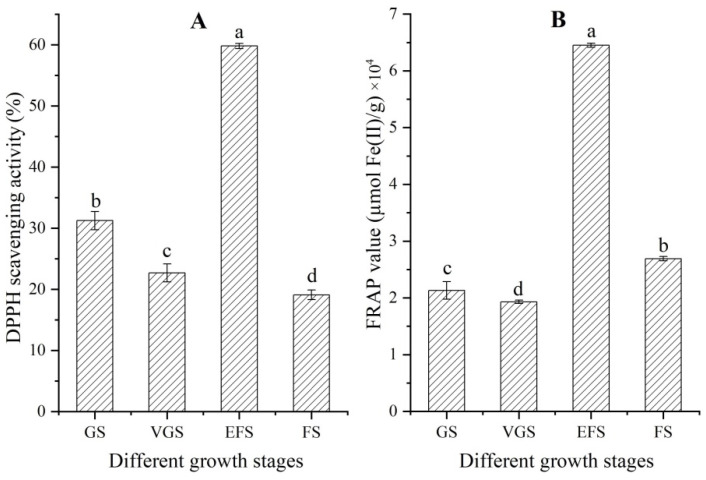
DPPH scavenging activity and FRAP value of extracts from stems of *C. songaricum* at four different growth stages (mean ± SD, n = 3). Images (**A**,**B**) represent DPPH scavenging activity and FRAP value, respectively. Different letters represent a significant difference (*p* < 0.05) at different growth stages.

**Figure 4 ijms-23-10675-f004:**
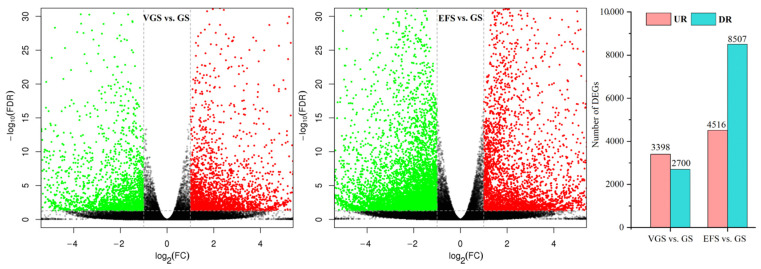
Volcano plot of unigenes and number of differentially expressed genes (DEGs) at VGS and EFS vs. GS.

**Figure 5 ijms-23-10675-f005:**
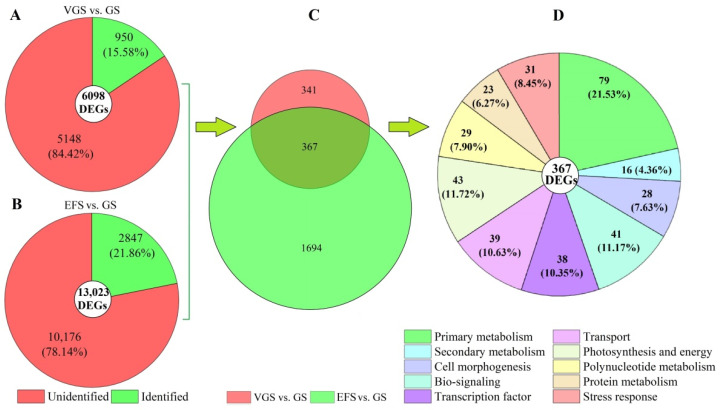
Distribution and classification of DEGs at VGS and EFS vs. GS. Images (**A**,**B**) represent the DEGs at VGS and EFS vs. GS, respectively; image (**C**) represents the co-expressed genes; image (**D**) represents the classification of the co-expressed genes.

**Figure 6 ijms-23-10675-f006:**
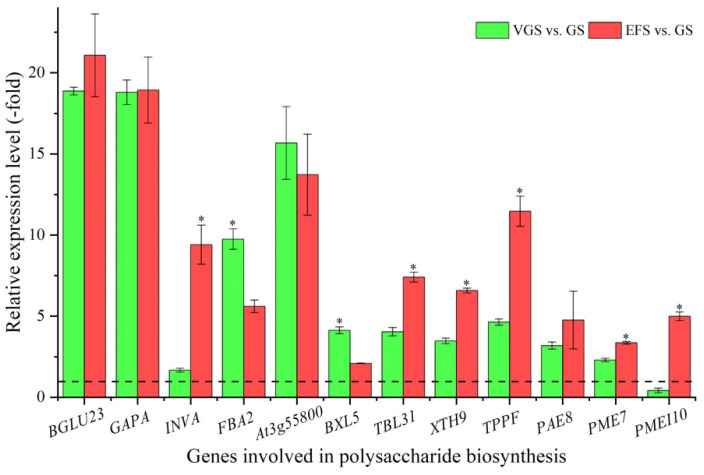
The relative expression level of genes involved in polysaccharide biosynthesis at VGS and EFS vs. GS, as determined by qRT-PCR (mean ± SD, n = 3). The “*” represents a significant difference (*p* < 0.05) at different growth stages for the same gene.

**Figure 7 ijms-23-10675-f007:**
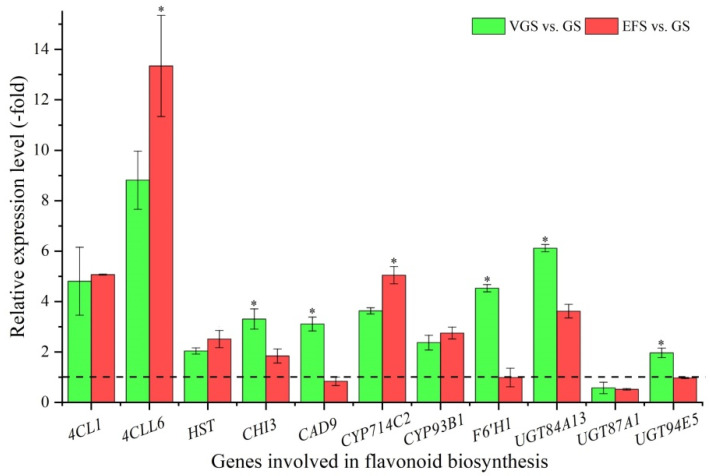
The relative expression level of genes involved in flavonoid biosynthesis at VGS and EFS vs. GS, as determined by qRT-PCR (mean ± SD, n = 3). The “*” represents a significant difference (*p* < 0.05) at different growth stages for the same gene.

**Figure 8 ijms-23-10675-f008:**
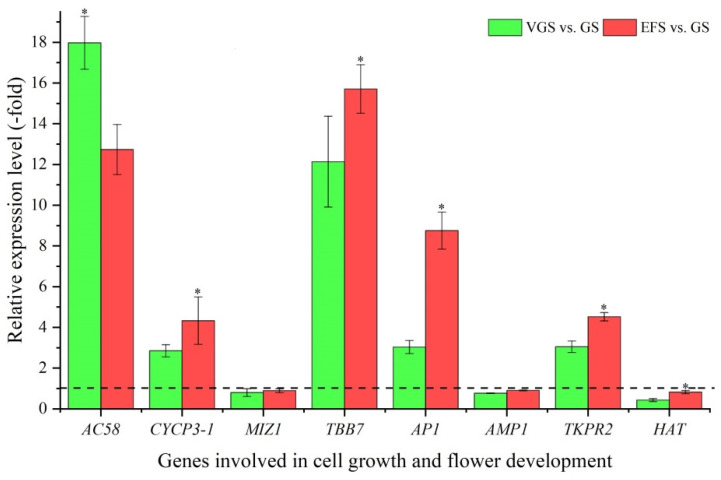
The relative expression level of genes involved in cell growth and flower development at VGS and EFS vs. GS, as determined by qRT-PCR (mean ± SD, n = 3). The “*” represents a significant difference (*p* < 0.05) at different growth stages for the same gene.

**Figure 9 ijms-23-10675-f009:**
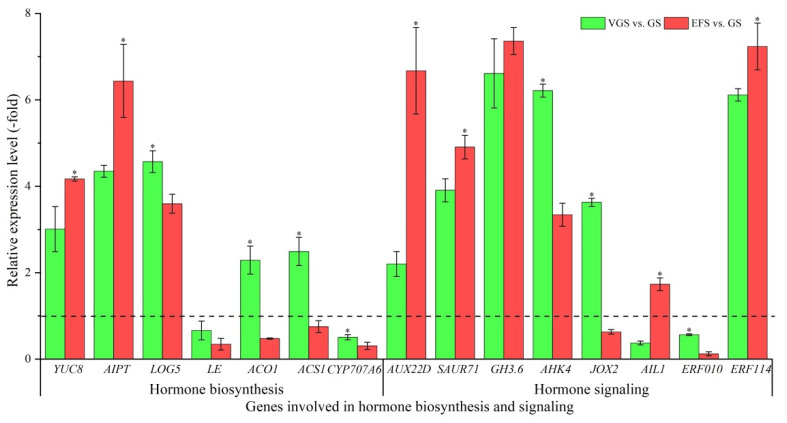
The relative expression level of genes involved in hormone biosynthesis and signaling at VGS and EFS vs. GS, as determined by qRT-PCR (mean ± SD, n = 3). The “*” represents a significant difference (*p* < 0.05) at different growth stages for the same gene.

**Figure 10 ijms-23-10675-f010:**
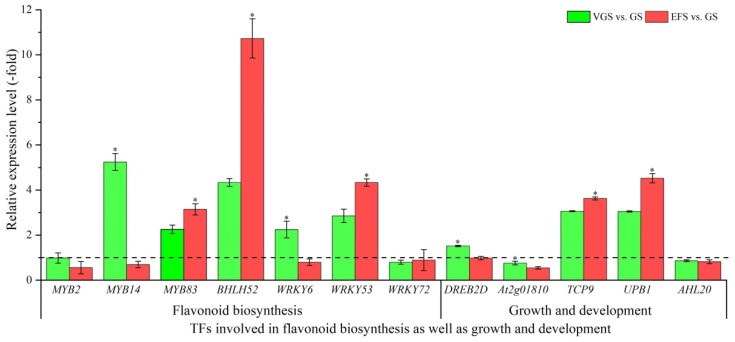
The relative expression level of TFs involved in flavonoid biosynthesis as well as growth and development at VGS and EFS vs. GS, as determined by qRT-PCR (mean ± SD, n = 3). The “*” represents a significant difference (*p* < 0.05) at different growth stages for the same gene.

**Figure 11 ijms-23-10675-f011:**
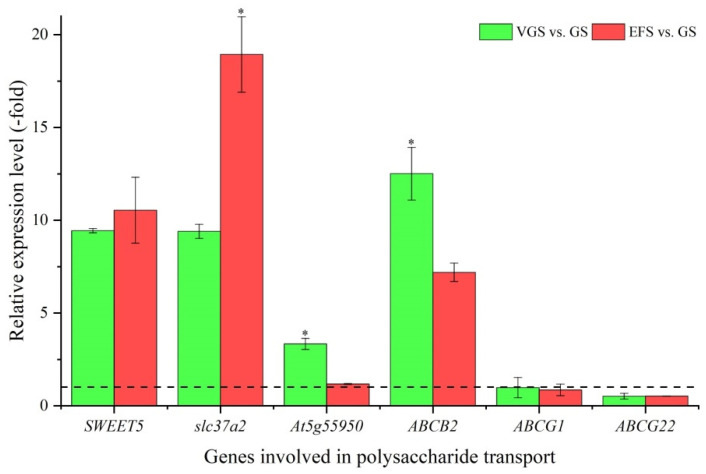
The relative expression level of genes involved in polysaccharide transport at VGS and EFS vs. GS, as determined by qRT-PCR (mean ± SD, n = 3). The “*” represents a significant difference (*p* < 0.05) at different growth stages for the same gene.

**Figure 12 ijms-23-10675-f012:**
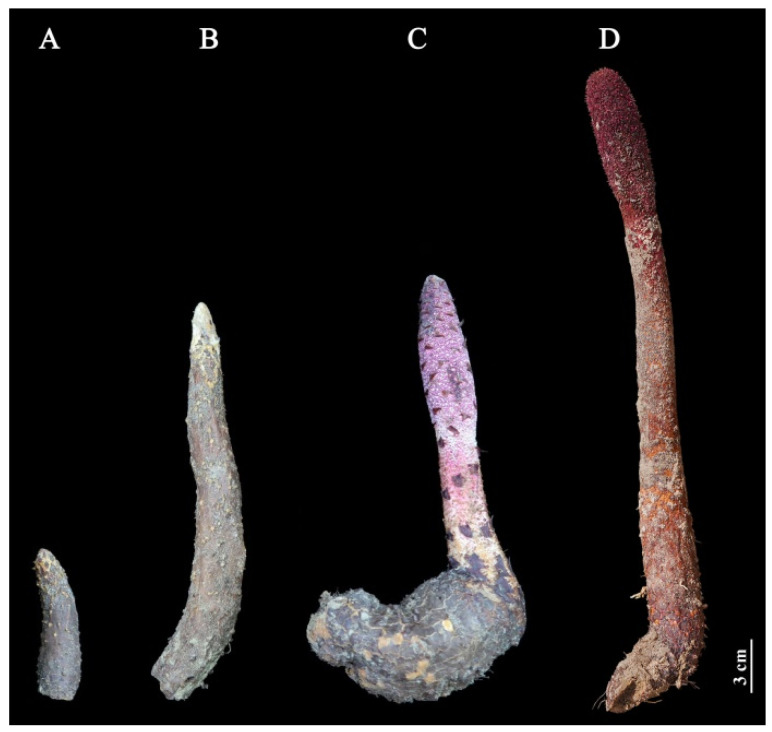
Morphological stem of *C. songaricum* at GS (**A**), VGS (**B**), EFS (**C**), and FS (**D**).

**Table 1 ijms-23-10675-t001:** Summary of sequencing data of *C. songaricum* at GS, VGS, and EFS (mean ± SD, n = 3).

	GS	VGS	EFS
Filtered data			
Data of reads number (million)	50.22 ± 5.50	51.24 ± 3.29	52.69 ± 3.95
Data of reads number × read length (million)	7533 ± 824.75	7686 ± 493.36	7904 ± 592.2
Q20 (%)	97.16 ± 0.19	96.83 ± 0.47	97.05 ± 0.27
Q30 (%)	92.21 ± 0.42	91.54 ± 0.92	92.00 ± 0.54
Mapped data			
Data of unique mapped reads (million)	41.17 ± 4.37	42.46 ± 2.79	43.59 ± 3.30
Data of multiple mapped reads (million)	1.62 ± 0.18	1.61 ± 0.13	1.62 ± 0.12
Mapping ratio (%)	85.21 ± 3.68	86.01 ± 4.22	85.80 ± 4.52

## Data Availability

The datasets are publicly available at NCBI with BioProject: PRJNA598928 (https://dataview.ncbi.nlm.nih.gov/object/PRJNA598928, accessed on 1 February 2021), and Sequence Read Archive (SRA) accession: GS (SRR10829656, SRR10829665 and SRR10829666), VGS (SRR10829653, SRR10829654 and SRR10829655), and EFS (SRR10829650, SRR10829651 and SRR10829652).

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
