# Peer review of "Integrated Metabolites and Transcriptomics at Different Growth Stages Reveal Polysaccharide and Flavonoid Biosynthesis in Cynomorium songaricum"

_ijms, 2022, doi:10.3390/ijms231810675_

Round 1

Reviewer 1 Report

The authors are conducting a study on Cynomorium songaricum (biomass taken from the natural environment), which is a perennial parasitic herb whose stem is used in traditional Chinese medicine for its richness in bioactive compounds (polysaccharides, flavonoids and triterpenes). Little information exists on the mechanism of accumulation of bioactives during the different stages of growth, the study carried out consists in characterizing the bioactive compounds (content of soluble sugar and flavonoids and antioxidant capacity) at 4 different stages of growth and the transcriptomics have been analyzed by sequencing and the expression levels of key genes are validated.
The numerous results fully demonstrate that the studied stem biomass, soluble sugars and total flavonoids content and antioxidant capacity peak at EFS compared to GS, VGS and FS. The results demonstrate that the accumulation of polysaccharides and flavonoids reaches the highest levels at EFS during growth phases. A total of 6098 and 13023 DEGs were observed at VGS and EFS vs. GS, respectively, with 109 genes directly integrated into involved in the biosynthesis of polysaccharides and flavonoids as well as in growth and development. These results therefore provide theoretical and useful information for the collection of C. songaricum, and the specific roles of key genes in the regulation of polysaccharides and flavonoids whose biosynthesis will require additional studies that the authors should specify.

The authors collect their biomass from the natural environment; aren't the results distorted by the fact that climatic conditions can impact the production of bioactives?

Why not conduct a parallel study on biomass from greenhouse cultivation?

In the study, the entire stem is used without distinction, but is there no difference according to the analyzed part (in particular at the level of the scaly part)?

Although the study is complete and the results allow fully substantiated conclusions, the introduction can be improved to better position the study carried out in relation to the numerous publications in the field.

As already specified above, the authors must better specify their perspectives in the conclusions of this study.

Author Response

1. These results therefore provide theoretical and useful information for the collection of C. songaricum, and the specific roles of key genes in the regulation of polysaccharides and flavonoids whose biosynthesis will require additional studies that the authors should specify.

Thanks for your suggestion, the key genes that will be further studied in the near future have been specifically described in the Conclusion section: “The specific roles of key genes in the regulation of polysaccharide (e.g., GLCs, XTHs and PMEs) and flavonoid (e.g., 4CLLs, CYPs and UGTs) biosynthesis will require additional studies.” (Page 14, lines 470-472)

2. The authors collect their biomass from the natural environment; aren't the results distorted by the fact that climatic conditions can impact the production of bioactives?

3. Why not conduct a parallel study on biomass from greenhouse cultivation?

Thanks for your valuable comments (Q2 and Q3), indeed, the production of bioactive compounds will affected by the climatic conditions in the natural environment. Currently, the Cynomorium songaricum plants are host in Nitraia species, which are wildly grown in the arid desert and saline-alkaline regions (See the followed Fig. 1). Thus, it is difficult to control the growth conditions (e.g., temperature, irrigation and light) during the whole growth and development stages. In this study, we focused on the changes of bioactive compounds in C. songaricum at 4 different growth stages. Your comments on the effect of climate conditions on bioactive compounds provide a new direction for our researches.

In addition, to our knowledge, there is no experiment conducted on the greenhouse cultivation of Cynomorium songaricum in the natural harsh environment conditions such as strong wind and sandstorm.

Fig. 1 has been attached in the PDF document.

Fig. 1 Growth conditions of C. songaricum plants host in Nitraia species in the arid desert regions.

4. In the study, the entire stem is used without distinction, but is there no difference according to the analyzed part (in particular at the level of the scaly part)?

In this study, the parts that were used in the experiments have been mentioned in the section of Materials and Methods: “the middle part of stem was used for determination of soluble sugar and total flavonoids contents as well as antioxidant capacity, and the shoot apical meristems (SAM) were used for transcriptomic analysis.” (Page 12, lines 382-385)

5. Although the study is complete and the results allow fully substantiated conclusions, the introduction can be improved to better position the study carried out in relation to the numerous publications in the field.

Thanks for your suggestion, the references directly involved in growth and bioactive compounds of C. songaricum have been added in the text: “The types and contents of liposoluble components (e.g., β-sitosterol, hexadecanoic acid and (Z)-9-octadecenoic acid) in C. songaricum were significantly different among different hosts [11]” and “At the molecular level to reveal the genes regulating bioactive compounds, two genes leucoanthocyanidin reductase (LAR) and dihydroflavonol 4-reductase (DFR) have been identified to be involved in catechin biosynthesis based on transcriptome analysis of C. Songaricum [18].”. (Page 2, lines 49-51 and lines 64-67)

    To date, there are limited researches and published literatures about the effect of environment conditions and growth stages on growth and metabolite accumulation of C. Songaricum, most of studies focused on the isolation of bioactive compounds and pharmacological efficacy.

6. As already specified above, the authors must better specify their perspectives in the conclusions of this study.

According to your comments, the perspectives have been specified in the section of Conclusions: “These findings will provide theoretical and useful information for the large-scale cultivation and collection of C. songaricum, as well as improve the yield and quality of C. songaricum by regulating the nutrient transport of the host Nitraia species.” (Page 14, lines 472-474).

Reviewer 2 Report

The authors compared the metabolites in the different growth stages of Cynomorium songaricum with the aid of biochemical studies and transcriptomics. The authors did a good experimental work. However, the result presentation can be improved.

1.     In section 2.1, How the stem fresh weight was calculated? Fresh weight per stem:  Equal length of the stem was taken or any other way was used to normalise the experiment? Mention in the methodology section.

2.     In section 2.2, and in 2.3, what is the need of analysing in per stem basis? Is the length of the stem taken for analysis is equal?

3.     Taking per stem as a sample may not be correct way. The length of each stem may vary for each growth level. Better to keep only the results based on biomass (fresh weight) might be scientifically correct way for any analysis.

4.     Table2 and Figure 6, Table 3 and Figure 7 and similar results which represents table and graph of the RT-PCR results for various conditions are not correlating. For example, in table 3, F’6H1 expression level (Log2FC) for EFS vs GS is -11.78. However, in the graph 7, fold change is shown near 1. As table and graphs are provided to give the same information, any one is enough.

5.     Check spelling (eg Change “includind: as “including” in line 77). Do through out the manuscript.

Author Response

1. In section 2.1, How the stem fresh weight was calculated? Fresh weight per stem:  Equal length of the stem was taken or any other way was used to normalise the experiment? Mention in the methodology section.

Thanks very much for your comments, in this study, the fresh weight per stem was taken based on the equal length of the stem, which has been added in the section of Materials and Methods: “Specifically, 10 sites of C. songaricum were randomly selected from 10 host plants of N. roborowskii, one-medium-length stem was chosen from the independent site of C. songaricum and measured using an electronic balance, and then the average FW of the stem was calculated. Generally, the C. songaricum plants grow in clusters and the size of stem is basically the same (Figure S9)”. (Page 12, lines 390-394)

Figure S9 has been attached in the PDF document.

Figure S9. Growth characteristics of C. songaricum between the VGS and EFS.

2. In section 2.2, and in 2.3, what is the need of analyzing in per stem basis? Is the length of the stem taken for analysis is equal?

3. Taking per stem as a sample may not be correct way. The length of each stem may vary for each growth level. Better to keep only the results based on biomass (fresh weight) might be scientifically correct way for any analysis.

According to your comments and suggestion, we plan to answer the Q2 and Q3 together.

Firstly, as mentioned in Q1, the length of stem that were taken for analysis is equal.

Secondly, according to your comments, the contents of soluble sugar and total flavonoids as well as antioxidant capacity have been shown based on the fresh weight. (Page 4, Figures 2 and 3).

4. Table 2 and Figure 6, Table 3 and Figure 7 and similar results which represents table and graph of the RT-PCR results for various conditions are not correlating. For example, in table 3, F’6H1 expression level (Log2FC) for EFS vs GS is -11.78. However, in the graph 7, fold change is shown near 1. As table and graphs are provided to give the same information, any one is enough.

Thanks for your suggestion, the Tables 2 to 7 that provided the same information as the Figures 6 to 11 have been provided as Supplemental materials.

Indeed, it is different between log2FC (syn. Reads Per kb per Million, RPKM) and 2-△△Ct (syn. relative expression level, REL). For the log2FC, the minus value shows negative expression; while for the 2-△△Ct, the 0 to 1 value shows negative expression and the >1 value shows positive expression. In fact, the expression level of F’6H1 expressed by log2FC=-11.78 is consistent with the level expressed by 2-△△Ct = 0.99-fold, both of which showed the negative expression of F’6H1 at EFS vs GS.

5. Check spelling (e.g., Change “includind: as “including” in line 77). Do throughout the manuscript.

We have tried our best to carefully check the spelling and avoid the errors throughout the text.

Round 2

Reviewer 2 Report

The authors significantly improved the manuscript. However, the manuscript needs few more corrections.

1.     Statistical significance is missing in all relative expression graphs (Figure 6 to Figure 11).

2.     In section 4.5, “The primer sequence (Table S17) was designed via a primer-blast in NCBI and synthesized by reverse transcription (Sangon Biotech Co., Ltd., Shanghai, China).” How primer synthesized by reverse transcription? Check and rewrite.

3.     Language correction: Change “Among of the” in line no 182 to “Among the”. Such corrections need to be checked throughout the manuscript.

Author Response

1. Statistical significance is missing in all relative expression graphs (Figure 6 to Figure 11).

According to your comments, the statistical significance analysis has been added in the Figures 6 to 11, and the description: “The “*” represents a significant difference (P < 0.05) at different growth stages for the same gene” has been also added in the titles. (Page 7, lines 213-214; Page 8, lines 227-228; Page 8, lines 242-243; Page 9, lines 259-260; Page 9, lines 277-278; Page 10, lines 290-291)

2. In section 4.5, “The primer sequence (Table S17) was designed via a primer-blast in NCBI and synthesized by reverse transcription (Sangon Biotech Co., Ltd., Shanghai, China).” How primer synthesized by reverse transcription? Check and rewrite.

Thanks for your careful and kind reviewing, the primer synthesized has been corrected: “The primer sequence (Table S17) was designed via a primer-blast in NCBI and synthesized by Sangon Biotech Co., Ltd. (Shanghai, China)”. (Page 14, lines 465-466)

3. Language correction: Change “Among of the” in line no 182 to “Among the”. Such corrections need to be checked throughout the manuscript.

The description of “Among of the” has been changed to “Among the” throughout the manuscript. (Page 6, lines 186 and 200; Page 7, line 216; Page 8, lines 230 and 245; Page 9, line 262; Page 10, line 280)